# GuidedNet: Semi-Supervised Multi-Organ Segmentation via Labeled Data Guide Unlabeled Data

## ABSTRACT

Semi-supervised multi-organ medical image segmentation aids physicians in improving disease diagnosis and treatment planning and reduces the time and effort required for organ annotation. Existing state-of-the-art methods train the labeled data with ground truths and train the unlabeled data with pseudo-labels. However, the two training flows are separate, which does not reflect the interrelationship between labeled and unlabeled data. To address this issue, we propose a semi-supervised multi-organ segmentation method called GuidedNet, which leverages the knowledge from labeled data to guide the training of unlabeled data. The primary goals of this study are to improve the quality of pseudo-labels for unlabeled data and to enhance the network's learning capability for both small and complex organs. A key concept is that voxel features from labeled and unlabeled data that are close to each other in the feature space are more likely to belong to the same class. On this basis, a 3D Consistent Gaussian Mixture Model (3D-CGMM) is designed to leverage the feature distributions from labeled data to rectify the generated pseudo-labels. Furthermore, we introduce a Knowledge Transfer Cross Pseudo Supervision (KT-CPS) strategy, which leverages the prior knowledge obtained from the labeled data to guide the training of the unlabeled data, thereby improving the segmentation accuracy for both small and complex organs. Extensive experiments on two public datasets, FLARE22 and AMOS, demonstrated that GuidedNet is capable of achieving state-of-the-art performance.

## CCS CONCEPTS

• **Computing methodologies → Computer vision**.

## KEYWORDS

Semi-Supervised Learning, 3D Medical Image Segmentation, Abdominal Organs, Gaussian Mixture Model

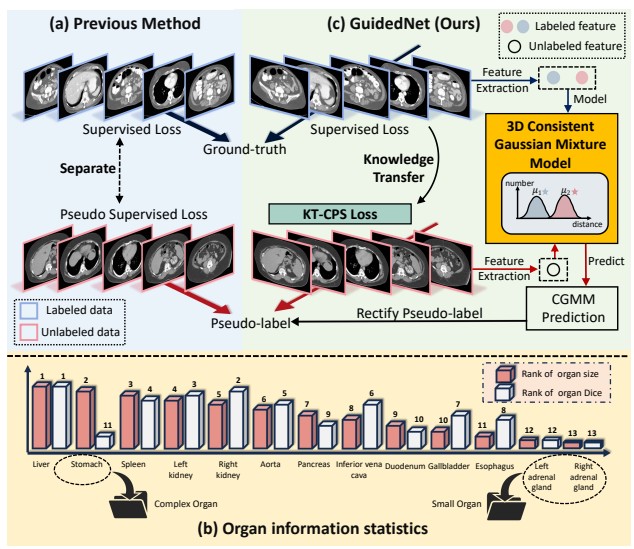

**Figure 1: (a) Previously developed pseudo-labeling methods separate the labeled and unlabeled data training flows, which does not reflect the interrelationship between labeled and unlabeled data (*i.e.*, CPS [4], ARCO [40], UCMT [30]). (b) In the FLARE22 dataset, rankings are provided for both the sizes of all foreground organs and the Dice. (c) Our GuidedNet, which is comprised of two components: 3D-CGMM and KT-CPS.**

## 1 INTRODUCTION

Accurately segmenting human organs from medical images is a crucial task that aids physicians in disease diagnosis, treatment planning, and follow-up care [44, 45]. However, precisely annotating medical images requires a significant amount of time, effort, and specialized knowledge [37]. In contrast, acquiring unlabeled images in clinical settings is much easier. To overcome the difficulty

in obtaining labeled images, researchers have attempted to utilize semi-supervised learning (SSL) [5, 18, 43], which entails training segmentation models using a small number of labeled images and a large number of unlabeled images.

The continuous development of SSL techniques has facilitated significant achievements in the field of organ segmentation. One important technique is pseudo-labeling [10], which generates pseudo-labels from unlabeled images for subsequent training [4, 14, 20, 30]. The most representative method is Cross Pseudo Supervision (CPS) [4], which imposes the consistency on two segmentation networks perturbed with different initialization for the same input image. Labeled data is supervised by ground truths in both segmentation networks, while unlabeled data from one perturbed segmentation network is supervised by pseudo-labels generated by the other segmentation network.

However, the processes employed to train the labeled and unlabeled data in these methods are separated, as depicted in Fig. 1(a). This separation causes a significant drawback: the quality of the pseudo-labels generated during training depends exclusively on the unlabeled data and the segmentation performance of the network, with no regard for the interrelationship between the labeled

and unlabeled data. This oversight leads to low-quality pseudo-labels, which adversely impacts the overall performance of the model. Additionally, widely used SSL multi-organ medical image segmentation methods focus on enhancing the learning ability of small organs due to the prevalent issue of class imbalance in medical image datasets. Specifically, the inherent complexity of organ segmentation must not be overlooked. For instance, the stomach, which is the second largest organ in abdominal multi-organ dataset, ranks only eleventh in terms of segmentation Dice similarity coefficient (referred to as Dice), as shown in Fig.1(b). This discrepancy highlights that the stomach's substantial size does not mitigate the segmentation difficulties posed by its complexity. Therefore, multi-organ segmentation is a complex task that requires careful consideration of the varying size and complexity levels of different organs.

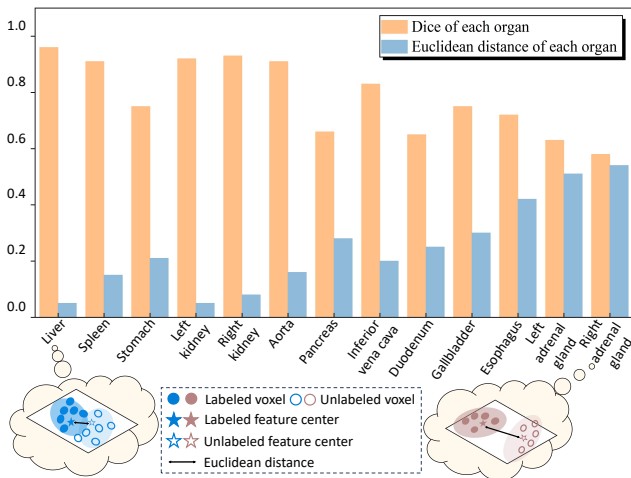

**Figure 2: Category-wise performance for the FLARE22 dataset. The orange bars represent the Dice for each organ, and the blue bars denote the Euclidean distances [7] between the feature centers of the labeled and unlabeled data for each organ.**

To address the above issue, we propose a novel semi-supervised multi-organ segmentation method named GuidedNet (as depicted in Fig.1(c)). The main objective of GuidedNet is to leverage the knowledge obtained from labeled data to guide the training of unlabeled data. To accomplish this, GuidedNet is designed with two main components: a 3D Consistent Gaussian Mixture Model (3D-CGMM) and Knowledge Transfer Cross Pseudo Supervision (KT-CPS) strategy. As illustrated in Fig.2, when a smaller Euclidean distance [7] exists between the feature centers of labeled and unlabeled data in a particular category, the Dice for that category is usually higher. Thus, when a high degree of similarity exists between the feature spaces of labeled and unlabeled voxels, the likelihood that they contain the same semantic information is high; hence, they should share their label. On this basis, we develop a 3D-CGMM that models the labeled data distribution of each class in the feature space using Gaussian mixture. During training, the 3D-CGMM is optimized using a training loss to ensure that it can

adapt in real-time. Subsequently, the 3D-CGMM is used to align the features of unlabeled voxels with the appropriate category-specific Gaussian mixture, which produces CGMM predictions. These predictions are then used to supervise segmentation predictions of the unlabeled data, leveraging guidance from the labeled data. This provides an additional training signal that assists in rectifying the pseudo-labels generated for the unlabeled data.

To address the challenging task due to differences in size and complexity among the different organs, we also implement the KT-CPS strategy, which measures organ learning challenges by comparing voxel predictions for the labeled data with the corresponding ground truth. Furthermore, we re-weight the pseudo supervised loss terms based on the learning challenges of different organs to overcome the segmentation difficulties posed by class imbalance and organ complexity.

The primary contributions of this study are as follows:

(1) We propose to leverage the knowledge from labeled data to rectify the generated pseudo-labels by using a 3D-CGMM that utilizes the feature distribution of the labeled data to generate CGMM predictions to guide the learning of unlabeled data.

(2) We design a KT-CPS strategy, which guides the training of unlabeled data to learn prior knowledge from the labeled data and enhances the learning ability of small and complex organs.

(3) Extensive experiments have been conducted to validate the effectiveness of our proposed GuidedNet. The results of these experiments have demonstrated solid performance gains across two public datasets.

## 2 RELATED WORK

### 2.1 Semi-Supervised Medical Image Segmentation

Semi-supervised segmentation methods applicable to medical images can be classified into types: consistency learning-based [2, 12, 31] and pseudo-labeling methods. One example of a consistency learning-based method is the mean teacher (MT) method [32], which employs both teacher and student models and enforces consistency loss based on perturbations in the unlabeled data. A number of studies refined the MT method by exploring various perturbation techniques and strategies [5, 25, 29]. In contrast, pseudo-labeling methods generate pseudo-labels for unlabeled data to guide the learning process[9, 19, 33]. The combination of consistency regularization and pseudo-labeling has become a leading approach for the semi-supervised semantic segmentation of medical images [26, 30]. Huang et al. [11] developed a novel convolutional neural network (CNN)-Transformer learning framework that effectively segmented medical images by producing complementary and reliable features and pseudo-labeling with bi-level uncertainty. Yuan et al. [42] proposed a new consistency regularization framework called mutual knowledge distillation (MKD), which was combined with data and feature augmentation. Recently, an increasing number of methods are focusing on enhancing the quality of pseudo-labels. For example, Zhao et al. [48] introduced rectified pseudo supervision to improve robustness under variant appearances in the image space. In addition, Chen et al. [5] utilized automatic clustering to model multiple prototypes, which helped alleviate the confirmation bias arising from noise and false labels.

**Figure 3: The workflow of GuidedNet involves processing input data from *model A* and *model B* to yield predictions. The feature distributions of the labeled predictions are utilized to train the 3D-CGMM, and the generated CGMM predictions are used to rectify the initial pseudo-labels. The prior knowledge obtained from the labeled predictions are transferred to the unlabeled predictions using the KT-CPS strategy for cross pseudo supervised training.**

## 2.2 Multi-Organ Segmentation

Accurately segmenting multiple organs is crucial for a variety of medical procedures [8]. Zhao et al. [47] proposed a lightweight network designed for multi-organ segmentation of abdominal computed tomography (CT) scans called LCOV-Net, which can segment 16 organs. LCOV-Net is a supervised network that relies on the quantity and quality of labeled data. Recently, a number of semi-supervised learning methods were developed for multi-organ segmentation in abdominal CT scans [23]. For example, Zhou et al. [49] proposed a deep multi-planar co-training (DMPCT) method that extracts consensus information from multiple views. In addition, Xia et al. [39] designed a Uncertainty-aware multi-view co-training (UMCT) method that corrected the prediction consistency between different scales and estimated the uncertainty of each view to generate reliable pseudo-labels. To mitigate the class-imbalance problem, Wang et al. [34] proposed a dual-debiased heterogeneous co-training (DHC) framework. Furthermore, Chen et al. [3] designed a data augmentation strategy (MagicNet) that corrected the original pseudo-label via cube-wise pseudo-label blending that incorporated crucial local attributes for identifying targets, especially small organs. The abovementioned semi-supervised learning methods improved the segmentation accuracy by utilizing unlabeled data; however, they failed to fully consider the underlying relationships between unlabeled and labeled images.

## 3 METHOD

### 3.1 Overview

An overview of the proposed GuidedNet is illustrated in Fig. 3, which is based on the CPS framework. This framework encompasses two parallel segmentation networks, *model A* and *model B*.

These networks share a common structure, but they are initialized differently; however, both are trained on the same input image. The purpose of this approach is to ensure that both networks produce consistent outputs. For the unlabeled data, each segmentation network can generate a pseudo-label, which serves as an additional signal to supervise the other segmentation network.

The training dataset, denoted as $\mathcal{S}$, is a union of labeled and unlabeled data. It is represented as $\mathcal{S} = \mathcal{S}^l \cup \mathcal{S}^u$, where $\mathcal{S}^l = \{(x_i, y_i)\}_{i=1}^N$ represents the labeled data, $\mathcal{S}^u = \{x_i\}_{i=1}^M$ represents the unlabeled data, and $N$ and $M$ indicate the number of labeled and unlabeled data, respectively ($M \gg N$ in most cases). In this context, $x_i \in \mathbb{R}^{D \times H \times W}$ denotes the input volume and $y_i \in \mathbb{R}^{K \times D \times H \times W}$ denotes the ground truth mask, where $K$ is the number of classes (including the background) and $H$, $W$, and $D$ represent the height, width, and depth of the input medical volume, respectively. The goal is to train a segmentation network based on $\mathcal{S}^l$ and $\mathcal{S}^u$ that correctly predicts labels for unseen data.

At each training step, labeled data (denoted as $\mathcal{B}_l$) and unlabeled data (denoted as $\mathcal{B}_u$) are sampled and fed into *model A* and *model B*, respectively. The supervised loss function is applied to the labeled data, guiding each segmentation head to generate a prediction mask that closely aligns with the ground-truth mask via

$$\mathcal{L}_{sup} = \frac{1}{|\mathcal{B}_l|} \sum_{i=1}^{\mathcal{B}_l} \left[ \mathcal{L}_s(p_i^A, y_i) + \mathcal{L}_s(p_i^B, y_i) \right], \quad (1)$$

where $\mathcal{L}_s = \frac{1}{2}[\mathcal{L}_{Dice} + \mathcal{L}_{ce}]$; $\mathcal{L}_{Dice}$ and $\mathcal{L}_{ce}$ represent the Dice and cross-entropy losses, respectively; and $p_i^{(\cdot)}$ is the output probability map. For all data, the KT-CPS loss and the CGMM loss are employed.

The optimization target is to minimize the overall loss, which can be formulated as

$$\mathcal{L} = \mathcal{L}_{sup} + \lambda_u \mathcal{L}_{kt-cps} + \lambda_g \mathcal{L}_{cgmm}, \tag{2}$$

where $\mathcal{L}_{sup}$, $\mathcal{L}_{kt-cps}$, and $\mathcal{L}_{cgmm}$ denote the supervised, KT-CPS, and CGMM losses, respectively, and $\lambda_u$ and $\lambda_g$ are the hyperparameters that can adjust the influence of each loss component within the model. We empirically set $\lambda_u$ to 0.1 and used the CPS weight ramp-up function [41] to gradually enlarge the $\mathcal{L}_{kt-cps}$ ratio. The value of $\lambda_g$ is set to 0.3. The training procedure utilized by GuidedNet is presented in Algorithm 1.

---

**Algorithm 1** Training Procedure of GuidedNet

---

**Input:** labeled data:$\mathcal{S}^l = \{(x_i, y_i)\}_{i=1}^N$, unlabeled data:$\mathcal{S}^u = \{x_i^u\}_{i=1}^M$,
    batchsize:$\mathcal{B}$, number of classes:$\textbf{\textit{num\_class}}$, max epoch:$E_{max}$
**Output:** Trained weights of $\textbf{\textit{model A}} f(\cdot; \theta_A)$ and $\textbf{\textit{model B}} f(\cdot; \theta_B)$
1: **for** epoch in $E_{max}$ **do**
2:     **for** batch $\mathcal{B}$ **do**
3:         Get predictions $p_A \leftarrow (x_i), p_B \leftarrow (x_i)$
4:         Get features $f_A \leftarrow (x_i), f_B \leftarrow (x_i)$
5:         **for** $x_i$ in $\mathcal{S}^l$ **do**
6:             Calculate $\mathcal{L}_{sup}$ according to Eqs.1
7:             **for** $j$ in num\_class $K$ **do**
8:                 Update $\mu_j, \sigma_j, \leftarrow (f_A, f_B)$
9:             **end for**
10:            Calculate $\mathcal{L}_{train}$ according to Eqs.12
11:         **end for**
12:         **for** $x_i$ in $\mathcal{S}^u$ **do**
13:            Get CGMM predictions $G'_A \leftarrow (x_i), G'_B \leftarrow (x_i)$
14:            Calculate $\mathcal{L}_{rectify}$ according to Eqs.13
15:         **end for**
16:         **for** $x_i$ in $\mathcal{S}^l \cup \mathcal{S}^u$ **do**
17:            Get weights $\omega_t^A$ and $\omega_t^B$ according to Eqs.18
18:            Calculate $\mathcal{L}_{kt-cps}$ according to Eqs.19
19:         **end for**
20:         Update $\mathcal{L}$ according to Eqs.2
21:         epoch = epoch + 1
22:     **end for**
23: **end for**return $model\ A f(\cdot; \theta_A)$ and $model\ B f(\cdot; \theta_B)$

---

## 3.2 3D Consistent Gaussian Mixture Model

In existing SSL segmentation methods, the training flows for labeled and unlabeled data are entirely separate, which overlooks the interrelationship between them. This oversight may lead to errors when generating pseudo-labels.

To rectify the generation of pseudo-labels for unlabeled data and guide the prediction process for unlabeled data, we introduce a 3D-CGMM that models the feature distribution of each class from labeled data. This approach ensures that the predictions for the unlabeled data do not solely depend on their own data but are also guided by the knowledge encapsulated in the labeled data, thereby enhancing the reliability and accuracy of the pseudo-labels.

**3D-CGMM Formulation.** Given input data with $K$ annotated classes, a 3D-CGMM with $K$ Gaussian mixtures is constructed. The center of the features from different categories are regarded as the centroids of the distinct Gaussian mixtures. For the $k_{th}$ Gaussian mixture, the mean features $\mu_k$ of labeled voxels $x_i$ belonging to the $k_{th}$ class are initially computed, as

represented by

$$\mu_k = \frac{1}{|\mathcal{B}_l|} \sum_{i=1}^{\mathcal{B}_l} f(x_i), \tag{3}$$

where $f(x_i)$ are the deep features of labeled voxels $x_i$, which are produced using the features before the classification layer of the segmentation model. After obtaining $\mu_k$, the variance $\sigma_k$ of the $k_{th}$ Gaussian mixture can be calculated as

$$\sigma_k = \sqrt{\frac{1}{|P_k|} \sum_{\forall x \in x_i} P_k (f(x) - \mu_k)^2}, \tag{4}$$

where $P_k$ denotes the segmentation prediction scores of the $k_{th}$ category.

Each voxel $x_i$ follows the probability density of the $k_{th}$ Gaussian mixture, which is calculated using the Gaussian probability density function:

$$\mathcal{N}(x_i \mid \mu_k, \sigma_k) = \frac{1}{\sqrt{2\pi}\sigma_k} \exp\left(-\frac{1}{2}(x_i - \mu_k)^2 \sigma_k^{-2}\right). \tag{5}$$

Then, following Bayes' rule [16], the posterior is derived as

$$P(x_i \mid k) = \frac{\pi_k \cdot \mathcal{N}(x|\mu_k, \Sigma_k)}{\sum_{i=1}^K \pi_i \cdot \mathcal{N}(x|\mu_i, \Sigma_i)}, \tag{6}$$

where $P(x_i \mid k)$ denotes the posterior probability that voxel $x_i$ belongs to the $k_{th}$ Gaussian mixture, $\pi_i$ denotes the mixing coefficient defined as $\frac{1}{K}$. The CGMM predictions $G$ are then produced using:

$$G_i = \sum_k^K P(x_i \mid k). \tag{7}$$

**3D-CGMM Training Loss.** Previous methods suggest using Expectation-Maximization (EM) algorithms [16, 27, 36] to formulate the Gaussian Mixture Model (GMM). However, this approach requires prior estimates and iterative updates of the parameters. In SSL, labeled voxels with available labels can serve as precise prior information useful for formulating the GMM. To train a 3D-CGMM effectively, selecting only voxels that have labels as reliable sources of information is essential.

In this paper, instead of using time-consuming EM algorithms, we leverage the reliable information of labeled data and employ an effective function $\mathcal{L}_{train}$ to adaptively optimize the 3D-CGMM. The 3D-CGMM training loss $\mathcal{L}_{train}$ contains four parts: a self-supervision loss $\mathcal{L}_{self}$, a ground truth loss $\mathcal{L}_{gt}$, a maximization loss $\mathcal{L}_{max}$, and a consistency loss $\mathcal{L}_{cons}$. First, the ground truth mask $y$ is assigned to supervise $G$ according to:

$$\mathcal{L}_{gt} = -\frac{1}{|\mathcal{B}_l|} \sum_{i=1}^{\mathcal{B}_l} y_i log(G_i). \tag{8}$$

Second, the CGMM prediction $G$ and output probability map $P$ of the model are utilized for self-supervision, employing a cross-entropy-based function to compute the loss $\mathcal{L}_{self}$, which is expressed as

$$\mathcal{L}_{self} = -\frac{1}{|\mathcal{B}_l|} \sum_{i=1}^{\mathcal{B}_l} [G_i * log(P_i) + (1 - G_i) * log(1 - P_i)]. \tag{9}$$

Third, for the purpose of learning discriminative Gaussian mixtures, a maximization loss $\mathcal{L}_{max}$ is used to enlarge the distance between the centroids of different Gaussian mixtures:

$$\mathcal{L}_{max} = \frac{2}{K(K-1)} \sum_{\forall k, v \in K, k \neq v} e^{-(\mu_k - \mu_v)^2}. \tag{10}$$

Additionally, in the semi-supervised framework, both *model A* and *model B* generate predictions for the unlabeled data, which are fed into the 3D-CGMM. The aim for the 3D-CGMM is to maintain consistency between the CGMM predictions $G'^A$ and $G'^B$ from *model A* and *model B*, respectively, as much as possible. Therefore, we employ the mean squared error (MSE) loss [35], which is expressed as

$$\mathcal{L}_{cons} = \frac{1}{|\mathcal{B}_u|} \sum_{i=1}^{\mathcal{B}_u} \left(G'^A - G'^B\right)^2. \tag{11}$$

The total loss function $\mathcal{L}_{train}$ for training the 3D-CGMM is expressed as

$$\mathcal{L}_{train} = \mathcal{L}_{self} + \mathcal{L}_{gt} + \mathcal{L}_{max} + \lambda_c \mathcal{L}_{cons}, \qquad (12)$$

where $\lambda_c$ is a hyperparameter used to balance the intensity of $\mathcal{L}_{cons}$.

**Rectify Pseudo-labels.** The 3D-CGMM simultaneously generates CGMM predictions $G'$ for the unlabeled data during the process of training the 3D-CGMM. Thus, $G'$ serves as a supervisory signal for the initial predictions of the unlabeled data, thereby facilitating the refinement of the pseudo-labels under the guidance of the labeled data. This methodology leverages the intrinsic data distribution captured by the 3D-CGMM and harnesses the predictive power of $G'$ to optimize the pseudo-labels, integrating the reliability of the labeled data to inform the processing of the unlabeled data via

$$\mathcal{L}_{rectify} = \frac{1}{|\mathcal{B}_u|} \sum_{i=1}^{\mathcal{B}_u} \left[ \mathcal{L}_{ce}(p_i^A, G_i'^A) + \mathcal{L}_s(p_i^B, G_i'^B) \right]. \qquad (13)$$

**Overall Loss.** The total loss function $\mathcal{L}_{cgmm}$ for the 3D-CGMM is expressed as

$$\mathcal{L}_{cgmm} = \mathcal{L}_{train} + \mathcal{L}_{rectify}. \qquad (14)$$

## 3.3 Knowledge Transfer Cross Pseudo Supervision Strategy

As mentioned in the previous section, significant disparities in organ sizes have been observed within multi-organ datasets containing labeled data. Organ size disparities constitute one of many aspects that need to be considered in organ segmentation. Many complexities associated with this process also need to be taken into account. Therefore, organs that are difficult to segment remain the primary focus of this study.

To address this challenge, we propose a strategy known as KT-CPS. This approach, which is guided by the prior knowledge obtained from the labeled data, forces the model to pay more attention to the organs that are either small or challenging to segment, thereby rebalancing the learning process for unlabeled data.

We denote the softmax probabilities of the $j_{th}$ voxel of the $i_{th}$ data generated by *model A* and *model B* as $\mathbf{p}_{ij}^A \in \mathbb{R}^K$ and $\mathbf{p}_{ij}^B \in \mathbb{R}^K$, respectively, where $K$ is the number of classes (including the background). Specifically, pseudo-labels are generated via

$$\hat{y}_{ij}^A = \arg\max_k p_{ij}^A(k), \ \hat{y}_{ij}^B = \arg\max_k p_{ij}^B(k), \qquad (15)$$

where $\hat{y}_{ij}^A$ and $\hat{y}_{ij}^B$ are the pseudo-labels of the $j_{th}$ voxel of the $i_{th}$ data given by *model A* and *model B*, respectively, and $p_{ij}^A(k)$ and $p_{ij}^B(k)$ are the $\mathbf{p}_{ij}^A$ and $\mathbf{p}_{ij}^B$ values for the $k_{th}$ dimension, respectively. The predicted categories from the model are compared to the true categories within a batch of labeled data using

$$\mathcal{R}_k^A = \frac{N_k^A}{\max\left\{N_k^A\right\}_{k=0}^K}, \mathcal{R}_k^B = \frac{N_k^B}{\max\left\{N_k^B\right\}_{k=0}^K}, \quad k = 0, 1, \dots, K, \qquad (16)$$

where $N_k^{(\cdot)}$ denotes the voxel number of the $k_{th}$ organ in labeled images in which the predicted category matches the actual label category. Based on the voxel proportions, the weight of each organ is calculated via:

$$w_k^A = \frac{\max\left\{\log\left(\mathcal{R}_k^A\right)\right\}_{k=0}^K}{\log\left(\mathcal{R}_k^A\right)}, w_k^B = \frac{\max\left\{\log\left(\mathcal{R}_k^B\right)\right\}_{k=0}^K}{\log\left(\mathcal{R}_k^B\right)}, \quad k = 0, 1, \dots, K. \qquad (17)$$

During training, the exponential moving average (EMA) of these parameters is leveraged to enhance the prediction stability after each training round via

$$\begin{cases} \omega_t^A = m * \omega_{t-1}^A + (1-m) * \omega_t^A, & \omega_t^A = [w_1^A, w_2^A, \dots, w_K^A], \\ \omega_t^B = m * \omega_{t-1}^B + (1-m) * \omega_t^B, & \omega_t^B = [w_1^B, w_2^B, \dots, w_K^B], \end{cases} \qquad (18)$$

where $m$ is the momentum parameter, which is experimentally determined to be 0.99. Based on the designed weights, the KT-CPS loss is defined as

$$\mathcal{L}_{kt-cps} = \frac{1}{|\mathcal{B}|} \sum_{i=1}^{\mathcal{B}} \left[ \omega_t^B \mathcal{L}_s(\mathbf{p}_{ij}^A, \hat{y}_{ij}^B) + \omega_t^A \mathcal{L}_s(\mathbf{p}_{ij}^B, \hat{y}^A ij) \right], \qquad (19)$$

where $\mathcal{B} = \mathcal{B}_l \cup \mathcal{B}_u$ signifies that the batch $\mathcal{B}$ consists of both labeled ($\mathcal{B}_l$) and unlabeled ($\mathcal{B}_u$)data. This combined batch is utilized in the calculation of the KT-CPS loss.

# 4 EXPERIMENTS

## 4.1 Experimental Setup

**Datasets.** We evaluate GuidedNet using two classical abdominal multi-organ segmentation datasets: FLARE22[24] and AMOS[13]. The FLARE22 dataset consists of 13 classes of organs (with one background): the liver (Liv), spleen (Spl), pancreas (Pan), right kidney (R.kid), left kidney (L.kid), stomach (Sto), gallbladder (Gal), esophagus (Eso), aorta (Aor), inferior vena cava (IVC), right adrenal gland (RAG), left adrenal gland (LAG), and duodenum (Duo). The dataset also includes 2000 unlabeled 3D CT volumes. To partition the dataset, we divide the labeled images into training, validation, and test sets using a ratio of 6:2:2. In addition, we incorporate the unlabeled images into the training set, resulting in labeled data proportions of 50% (42 labeled cases and 42 unlabeled cases) and 10% (42 labeled cases and 378 unlabeled cases) within two training sets. The AMOS dataset is comprised of 300 CT images, which are annotated at the pixel level for 15 distinct abdominal organs, including two additional organs not found in the FLARE22 dataset: the bladder (Bl) and prostate/uterus (P/U). In our experiments, we divide the dataset into training, validation, and test sets using a ratio of 6:2:2. For the training set, the proportion of labeled data is set to 10% and 50%.

**Data Preprocessing.** For both datasets, the data are preprocessed before the network is trained. Specifically, the orientations of all the CT scans are standardized in the left-posterior-inferior (LPI) direction. Additionally, the following three preprocessing procedures are applied. (1) The voxel values are clipped to the range of [−325, 325] Hounsfield units (HU) to enhance the contrast of the foreground organs and suppress the background interference. (2) The voxel spacing is standardized to [1.25, 1.25, 2.5]. (3) Min-max normalization is implemented via $(x - x_{0.5})/(x_{99.5} - x_{0.5})$, where $x_{0.5}$ and $x_{99.5}$ represent the 0.5th and 99.5th percentiles of $x$, respectively.

The regions enclosed by the dashed yellow boxes indicate misclassification executed by the model; our method corrects these errors within these regions.

**Implementation Details.** Following the settings used in previous studies [34], multiple data augmentation methods, *i.e.*, random crop and random flip are adopted. The random crop size is set to $64 \times 128 \times 128$. The batch size is set to eight, comprising four labeled and four unlabeled data. For training, we employ the stochastic gradient descent (SGD) optimizer for training with a weight decay of 0.0001 and momentum of 0.9 [22]. For both the FLARE22 and AMOS datasets, all the methods are trained using 20000 iterations and an initial learning rate of 0.1. During the network training, a polynomial learning rate policy is employed during network training to decrease the learning rate according to the formula $(1 - \frac{iteration}{max\_iteration})^{0.9}$ [3], where $iteration$ and $max\_iteration$ denote the current iteration and the total number of iterations, respectively. During the inference process, the final volumetric segmentation is generated using a sliding-window strategy, with a stride of $32 \times 80 \times 80$ and the sliding-window approach employs a patch size of $64 \times 160 \times 160$. We conduct the experiments on Pytorch [28] with two NVIDIA A100 GPUs.

**Evaluation Metrics.** In our experiments, the segmentation performances of the different methods are evaluated using two standard evaluation metrics: the Dice and Jaccard indices (referred to as Jaccard). The Dice and Jaccard values range from 0 to 1, with higher scores indicating more accurate segmentation. To reduce the randomness of the network training,

**Table 1:** Quantitative results (mean Dice for each organ, mean and SD of Dice, and mean and SD of Jaccard) for different methods applied to the FLARE22 dataset. 'Sup only' denotes supervised 3D U-Net [6]. The **bold** and underlined text denote the best and second best performances, respectively.

| Methods | Labeled/Unlabeled | Liv | Spl | Sto | L.kid | R.kid | Aor | Pan | IVC | Duo | Gal | Eso | RAG | LAG | Mean Dice | Mean Jaccard |
|---|---|---|---|---|---|---|---|---|---|---|---|---|---|---|---|---|
| | | | | | | | Mean Dice for each organ | | | | | | | | | |
| Sup only | 42/0(100%) | 94.21 | 88.32 | 49.40 | 91.68 | 91.33 | 89.89 | 48.83 | 79.24 | 52.10 | 56.24 | 61.13 | 44.87 | 42.12 | 68.41 ± 0.58 | 56.97 ± 0.42 |
| DAN [46] [MICCAI'17] | 42/42(50%) | 96.50 | 89.74 | 62.04 | 93.63 | 93.24 | 90.76 | 61.71 | 80.09 | 66.60 | 70.07 | 67.20 | 58.04 | 43.47 | 74.86 ± 0.69 | 63.73 ± 0.62 |
| MT [32] [Neurips'17] | 42/42(50%) | 96.97 | 89.64 | 66.63 | 93.66 | 92.58 | 91.39 | 68.65 | 82.08 | 60.96 | 69.89 | 71.68 | 57.51 | 62.21 | 77.22 ± 0.42 | 65.94 ± 0.83 |
| UA-MT [41] [MICCAI'19] | 42/42(50%) | **97.21** | 88.85 | 71.66 | 94.00 | 93.50 | 92.41 | 70.60 | 82.92 | 64.85 | 76.82 | 72.05 | 60.02 | 60.85 | 78.91 ± 0.89 | 68.01 ± 1.21 |
| SASSnet [15] [MICCAI'20] | 42/42(50%) | 95.25 | 92.03 | 66.48 | 92.47 | **93.79** | 90.03 | 63.61 | 79.94 | 60.14 | 65.57 | 70.83 | 59.98 | 62.78 | 76.38 ± 0.61 | 65.20 ± 0.60 |
| DTC [21] [AAAI'21] | 42/42(50%) | 96.47 | 91.33 | 65.94 | **94.46** | 93.57 | **92.52** | 64.88 | 83.77 | 65.58 | 75.80 | 68.53 | 68.87 | 61.35 | 78.70 ± 0.79 | 67.64 ± 1.05 |
| CPS [4] [CVPR'21] | 42/42(50%) | 96.77 | 91.23 | 72.63 | 93.38 | 93.70 | 92.35 | 70.34 | 83.30 | 65.48 | 78.81 | 72.37 | 58.34 | 61.47 | 79.24 ± 0.56 | 67.99 ± 0.56 |
| CLD [17] [MICCAI'22] | 42/42(50%) | 94.27 | 88.54 | 74.88 | 91.48 | 93.33 | 91.51 | 71.51 | 83.21 | 68.15 | 76.68 | 71.03 | 64.57 | 63.98 | 79.47 ± 0.27 | 67.95 ± 0.28 |
| DHC [34] [MICCAI'23] | 42/42(50%) | 92.54 | 90.81 | 76.96 | 93.39 | 92.18 | 91.84 | 74.65 | 83.25 | 69.44 | 84.60 | 72.91 | 64.52 | 55.88 | 80.23 ± 1.07 | 68.90 ± 1.26 |
| MagicNet [3] [CVPR'23] | 42/42(50%) | 96.49 | 88.84 | 80.33 | 90.84 | 93.06 | 91.72 | 69.46 | 81.98 | 67.44 | 82.81 | 75.79 | 63.40 | 59.98 | 80.16 ± 0.33 | 69.08 ± 0.12 |
| **GuidedNet (Ours)** | 42/42(50%) | 96.09 | **92.10** | **80.60** | 93.45 | 93.64 | 92.44 | **75.73** | **84.28** | **71.15** | **85.26** | **76.09** | **71.34** | **68.33** | **83.12 ± 0.33** | **72.47 ± 0.38** |
| DAN [46] [MICCAI'17] | 42/378(10%) | 95.89 | 84.15 | 67.29 | 92.81 | 91.96 | 91.35 | 63.12 | 79.33 | 66.48 | 77.29 | 67.82 | 50.41 | 48.41 | 75.10 ± 0.69 | 63.83 ± 0.23 |
| MT [32] [Neurips'17] | 42/378(10%) | 96.49 | 91.54 | 74.64 | 93.78 | 92.77 | 92.17 | 69.23 | 82.71 | 66.68 | 73.49 | 70.66 | 61.88 | 41.26 | 77.49 ± 0.48 | 66.45 ± 0.39 |
| UA-MT [41] [MICCAI'19] | 42/378(10%) | 96.42 | 91.98 | 79.91 | 92.74 | 92.83 | 92.33 | 71.43 | 83.10 | 67.72 | 77.26 | 72.41 | 64.04 | 46.18 | 79.10 ± 0.38 | 68.21 ± 0.48 |
| SASSnet [15] [MICCAI'20] | 42/378(10%) | 96.21 | 90.40 | 67.12 | **94.00** | 92.85 | 91.61 | 67.89 | 79.59 | 65.47 | 71.59 | 71.44 | 52.07 | 57.83 | 76.77 ± 0.30 | 65.89 ± 0.44 |
| DTC [21] [AAAI'21] | 42/378(10%) | 96.63 | 92.91 | 72.76 | 92.68 | 92.40 | 91.87 | 66.82 | 81.47 | 65.76 | 78.38 | 69.39 | 59.74 | 59.10 | 78.45 ± 0.82 | 67.05 ± 1.02 |
| CPS [4] [CVPR'21] | 42/378(10%) | 96.62 | 92.16 | 77.02 | 92.70 | 92.71 | 92.25 | 69.39 | 81.91 | 65.94 | 75.12 | 72.78 | 63.56 | 58.96 | 79.32 ± 0.46 | 68.14 ± 0.61 |
| CLD [17] [MICCAI'22] | 42/378(10%) | 94.63 | 89.74 | 73.20 | 91.76 | 92.97 | 91.61 | 70.27 | 83.12 | 68.13 | 84.15 | 72.69 | 67.89 | 55.27 | 79.65 ± 0.17 | 68.22 ± 0.49 |
| DHC [34] [MICCAI'23] | 42/378(10%) | 93.17 | 90.64 | 80.56 | 93.13 | 92.89 | 91.38 | 72.22 | 83.75 | 69.73 | 82.47 | 73.25 | 67.12 | 56.19 | 80.50 ± 0.43 | 69.38 ± 0.63 |
| MagicNet [3] [CVPR'23] | 42/378(10%) | **97.04** | 88.04 | 81.51 | 92.18 | 92.95 | 91.75 | 71.15 | 81.01 | 69.61 | 84.36 | **77.07** | 63.34 | 60.33 | 80.79 ± 0.75 | 70.23 ± 0.96 |
| **GuidedNet (Ours)** | 42/378(10%) | 96.77 | **93.48** | **83.19** | **94.51** | **93.48** | **92.95** | **75.97** | **84.63** | **71.92** | **85.87** | 75.74 | **71.10** | **67.77** | **83.64 ± 0.42** | **73.08 ± 0.38** |

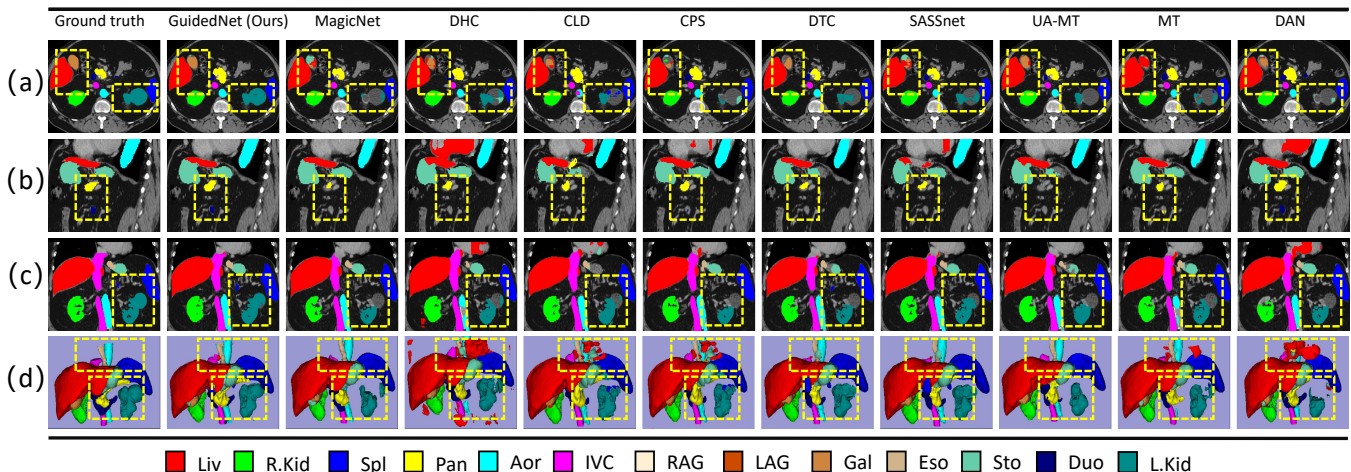

**Figure 4: Visualization of the segmentation results on the FLARE22 dataset. (a-d)Segmentation results of one case on transverse section, coronal section, sagittal section, and 3D view, respectively. The regions enclosed by the dashed yellow boxes indicate misclassification executed by the model; our method corrects these errors within these regions.**

experiments are calculated in triplicate for all methods and the mean and standard deviation (SD) of the Dice and Jaccard values are calculated. The model weights are determined based on the performance of the validation set, and the different methods are compared with the segmentation indices using the test set.

## 4.2 Comparison to SOTA Methods

To further assess the performance of GuidedNet, we compare our method to nine state-of-the-art semi-supervised segmentation methods: (1) deep adversarial networks (DAN) [46], (2) MT [32], (3) uncertainty-aware mean teacher (UA-MT) [41], (4) the shape-aware semi-supervised network (SASSnet) [15], (5) the dual-task consistency (DTC) framework [21], (6) CPS [4], (7) calibrating label distribution (CLD) for segmentation [17], (8) the dual-debiased heterogeneous co-training (DHC) framework [34], (9) MagicNet [3] and the

fully supervised 3D U-Net [6]. For all semi-supervised methods, we utilize 3D U-Net as the backbone. Comparison experiments are performed on the FLARE22 and AMOS datasets.

**FLARE22.** For the FLARE22 dataset, we train the semi-supervised models using training sets with labeled data proportions of 50% and 10%. The supervised 3D U-Net is trained using all 42 labeled data cases. The quantitative results obtained using the different methods are listed in Table.1. Compared to the supervised 3D U-Net, all semi-supervised methods achieve higher mean Dice and mean Jaccard by utilizing the unlabeled data. GuidedNet significantly outperforms all the other methods, achieving a superior state-of-the-art performance. With only 50% labeled data, its mean Dice is 82.13% and its mean Jaccard is 72.47%, surpassing other semi-supervised methods by 1.90%–7.27% and 3.39%–8.74%, respectively. GuidedNet also attains the highest mean Dice and mean Jaccard for the training set with a

**Table 2: Quantitative results (mean Dice for each organ, mean and SD of Dice, and mean and SD of Jaccard) for different methods applied to the AMOS dataset. 'Sup only' denotes supervised 3D U-Net [6]. The bold and underlined text denote the best and second best performances, respectively.**

| Methods | Labeled/Unlabeled | Liv | Sto | Spl | L.kid | R.kid | Aor | Bla | IVC | Pan | Duo | P/U | Gal | Eso | RAG | LAG | Mean Dice | Mean Jaccard |
|---|---|---|---|---|---|---|---|---|---|---|---|---|---|---|---|---|---|---|
| Sup only | 18/0(100%) | 85.99 | 40.08 | 82.19 | 79.57 | 80.24 | 75.51 | 11.63 | 59.86 | 26.80 | 20.22 | 22.40 | 15.06 | 37.05 | 32.46 | 9.68 | 45.24 ± 0.65 | 39.06 ± 0.42 |
| DAN [46] [MICCAI'17] | 18/162(10%) | 86.54 | 50.00 | 83.98 | 86.97 | 85.76 | 85.46 | 54.81 | 67.26 | 48.97 | 43.84 | 52.23 | 33.08 | 40.28 | 28.00 | 18.09 | 57.80 ± 0.88 | 47.38 ± 0.89 |
| MT [32] [Neurips'17] | 18/162(10%) | 89.26 | 56.87 | 84.27 | 84.42 | 85.98 | 85.57 | 51.22 | 70.13 | 48.91 | 48.04 | 39.46 | 42.17 | 50.71 | 43.92 | 30.46 | 61.44 ± 1.28 | 51.00 ± 1.10 |
| UA-MT [41] [MICCAI'19] | 18/162(10%) | 88.25 | 52.49 | 86.39 | 86.34 | 87.73 | 86.14 | 68.22 | 70.76 | 47.19 | 42.79 | 49.54 | 32.49 | 52.87 | 43.98 | 37.76 | 61.73 ± 1.12 | 50.97 ± 0.90 |
| SASSnet [15] [MICCAI'20] | 18/162(10%) | **90.33** | 48.61 | 86.87 | 87.66 | **88.17** | 87.09 | 43.55 | 73.85 | 50.29 | 48.56 | 8.38 | 36.15 | 43.18 | 41.36 | 28.85 | 58.35 ± 1.42 | 51.15 ± 0.83 |
| DTC [21] [AAAI'21] | 18/162(10%) | 89.81 | 50.49 | 87.48 | 85.20 | 85.84 | 85.83 | 64.49 | 72.72 | 43.44 | 47.36 | 39.19 | 38.62 | 50.56 | 42.53 | 37.02 | 60.81 ± 1.27 | 50.84 ± 1.24 |
| CPS [4] [CVPR'21] | 18/162(10%) | 88.52 | 55.52 | 83.25 | 86.30 | 87.97 | 85.36 | 60.53 | 71.71 | 50.11 | 46.05 | 60.33 | 60.33 | 52.37 | 46.33 | 37.48 | 63.52 ± 0.36 | 51.82 ± 0.49 |
| CLD [17] [MICCAI'22] | 18/162(10%) | 88.43 | 63.71 | 84.90 | 85.85 | 86.07 | 85.16 | 64.15 | 75.56 | 55.21 | 49.67 | 60.62 | 39.47 | 56.71 | 50.91 | 40.56 | 65.81 ± 1.24 | 54.00 ± 1.69 |
| DHC [34] [MICCAI'23] | 18/162(10%) | 83.27 | 63.39 | 83.60 | 84.11 | 85.66 | 84.40 | **74.52** | 74.88 | 56.02 | 51.89 | 65.47 | 47.53 | 43.21 | 48.28 | 42.59 | 65.17 ± 1.47 | 52.46 ± 1.30 |
| MagicNet [3] [CVPR'23] | 18/162(10%) | 88.99 | 61.20 | 83.52 | **88.39** | 87.24 | 83.69 | 62.47 | 74.83 | 53.04 | 52.88 | 57.28 | 56.69 | 55.68 | 46.87 | 43.16 | 65.31 ± 1.31 | 54.89 ± 0.78 |
| **GuidedNet (Ours)** | 18/162(10%) | 89.08 | **66.44** | **87.50** | 85.86 | 87.25 | **87.93** | 70.65 | **76.32** | **58.38** | **55.55** | **67.68** | 48.95 | **59.87** | **54.11** | **43.40** | **69.19 ± 0.17** | **56.97 ± 0.15** |
| Sup only | 90/0(100%) | 89.25 | 55.60 | 84.23 | 87.40 | 88.58 | 87.32 | 53.49 | 73.71 | 48.56 | 48.21 | 52.68 | 38.43 | 50.27 | 38.48 | 30.30 | 61.29 ± 1.74 | 51.62 ± 1.35 |
| DAN [46] [MICCAI'17] | 90/90(50%) | 90.49 | 55.91 | 89.63 | 90.08 | 88.74 | 86.71 | 47.44 | 72.09 | 54.98 | 50.33 | 53.04 | 39.13 | 58.34 | 29.57 | 6.49 | 61.39 ± 1.16 | 52.06 ± 1.45 |
| MT [32] [Neurips'17] | 90/90(50%) | 92.08 | 62.02 | 89.83 | 90.23 | 89.24 | 89.12 | 63.05 | 78.11 | 53.46 | 52.85 | 40.93 | 51.63 | 59.64 | 45.41 | 37.35 | 66.17 ± 0.75 | 57.06 ± 1.00 |
| UA-MT [41] [MICCAI'19] | 90/90(50%) | 90.86 | 58.55 | 88.92 | 88.93 | 88.83 | 88.49 | 54.86 | 74.28 | 51.88 | 54.54 | 44.73 | 40.99 | 58.58 | 51.31 | 41.78 | 65.48 ± 0.80 | 55.62 ± 1.10 |
| SASSnet [15] [MICCAI'20] | 90/90(50%) | 91.65 | 53.00 | 91.54 | 89.61 | 89.72 | 88.50 | 50.43 | 74.87 | 46.34 | 52.48 | 55.92 | 37.93 | 60.57 | 45.62 | 39.17 | 63.77 ± 1.13 | 54.68 ± 0.55 |
| DTC [21] [AAAI'21] | 90/90(50%) | 91.25 | 56.49 | 90.68 | 88.88 | 89.30 | 89.16 | 67.37 | 76.50 | 48.13 | 54.67 | 54.23 | 41.88 | 62.49 | 47.67 | 42.91 | 66.93 ± 1.78 | 55.92 ± 1.78 |
| CPS [4] [CVPR'21] | 90/90(50%) | 90.94 | 61.90 | 89.97 | 90.25 | 89.67 | 88.77 | 65.03 | 75.27 | 52.34 | 45.15 | 54.76 | 42.87 | 62.44 | 49.96 | 47.74 | 66.65 ± 1.24 | 56.56 ± 0.54 |
| CLD [17] [MICCAI'22] | 90/90(50%) | 91.23 | 66.18 | 89.34 | 89.50 | 89.86 | 88.85 | 66.40 | 76.97 | 55.63 | 53.35 | 58.82 | 45.78 | 62.93 | 54.24 | 43.79 | 69.09 ± 1.14 | 57.99 ± 1.14 |
| DHC [34] [MICCAI'23] | 90/90(50%) | 86.68 | 58.39 | 86.62 | 85.57 | 87.48 | 87.28 | 67.04 | 74.38 | 60.88 | 56.91 | 58.87 | 53.75 | 54.14 | 51.59 | 51.03 | 68.60 ± 0.56 | 56.05 ± 0.51 |
| MagicNet [3] [CVPR'23] | 90/90(50%) | 91.69 | 66.33 | 88.59 | 90.28 | 89.64 | 86.80 | 61.80 | 74.39 | 59.94 | 52.88 | 57.28 | 58.83 | 59.53 | 52.74 | 42.35 | 68.94 ± 0.56 | 58.33 ± 0.52 |
| **GuidedNet (Ours)** | 90/90(50%) | **92.32** | **72.99** | 91.21 | **90.82** | **89.87** | **89.31** | **74.00** | **78.41** | **61.75** | **58.26** | **65.23** | 56.72 | **66.62** | **54.71** | **51.10** | **72.94 ± 0.09** | **61.53 ± 0.12** |

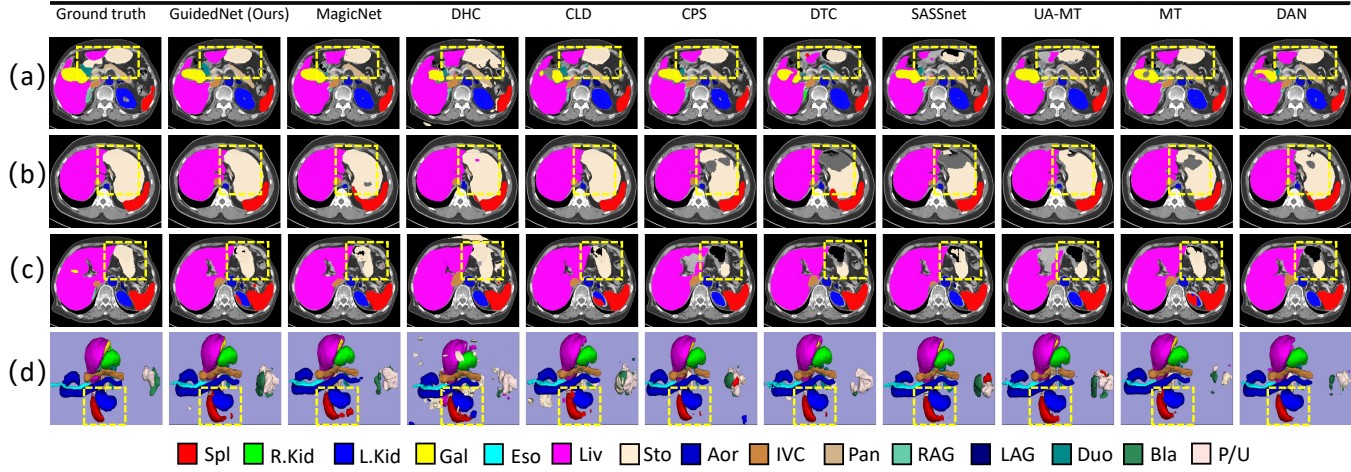

**Figure 5: Visualization of the segmentation results for the AMOS dataset. (a-c) Segmentation results for one case of three transverse sections and (d) 3D segmentation views. The regions enclosed by the dashed yellow boxes indicate misclassification executed by the model; our method corrects these errors within these regions.**

labeled data proportion of 10%. Additionally, GuidedNet exhibits excellent performance in terms of the mean Dice of large organs (*i.e.*, Liv, L.Kid, and R.kid), with significant improvements in the Dice of smaller organs (*i.e.*, Gal, RAG, and LAG), and complex organs (*i.e.*, Sto and Pan). In terms of Dice, the results for the smallest organ (*i.e.*, LAG) surpasses those of other semi-supervised methods by 7.44%–26.51%, and the results for the most complex organ (*i.e.*, Sto) also outperforms those of other semi-supervised methods by 1.68%–16.07%. These results validate the superiority of the 3D multi-organ segmentation performance of GuidedNet, particularly for small and complex organs. Fig.4 illustrates the qualitative results of different methods when using 378 unlabeled images from the FLARE22 dataset. The figure shows that GuidedNet accurately segments organs of various sizes and shapes. Furthermore, compared to other methods, GuidedNet generates clearer and more well-defined organ boundaries, mitigating issues with abnormal organ segmentation.

**AMOS.** To further validate GuidedNet, we conduct experiments on the AMOS dataset. The results demonstrate that our method outperforms limited supervision (Sup only) by a significant margin, with mean Dice improvements of 23.95% and 11.65% for 18 and 90 labeled samples, respectively. Additionally, our method has shown greater robustness compared to other semi-supervised methods on the AMOS dataset. For both training sets with 10% and 50% labeled data, CLD achieves a higher mean Dice, and MigicNet achieves a higher mean Jaccard than the other methods. Under 10% labeled data, our approach exhibits a 3.38% improvement in Dice and a 2.10% improvement in Jaccard compared to the current top-ranked method. Similarly, with 50% labeled data, our approach demonstrates a 3.85% improvement in Dice and a 3.20% improvement in Jaccard compared to the current top-ranked method. To provide a qualitative comparison, the visualization results for one test case of three transverse sections, along with 3D views, are shown in Fig. 5. Our approach is consistent with the quantitative

**Table 3: Quantitative results (mean Dice for each organ, mean and SD of Dice, and mean and SD of Jaccard) for the ablation study of the KT-CPS and 3D-CGMM. The bold and underlined text denote the best and second best performances, respectively.**

| Baseline | KT-CPS | 3D-CGMM | Mean Dice for each organ | | | | | | | | | | | | | Mean Dice | Mean Jaccard |
|---|---|---|---|---|---|---|---|---|---|---|---|---|---|---|---|---|---|
| | | | Liv | Spl | Sto | L.kid | R.kid | Aor | Pan | IVC | Duo | Gal | Eso | RAG | LAG | | |
| ✓ | | | 96.92 | 91.86 | 77.02 | 92.70 | 92.71 | 92.25 | 69.39 | 81.91 | 65.94 | 75.12 | 72.78 | 63.56 | 58.96 | 79.32 ± 0.46 | 68.14 ± 0.61 |
| ✓ | ✓ | | 96.71 | 93.12 | 81.53 | 94.47 | 92.96 | 92.62 | 73.53 | 83.98 | 70.43 | 75.61 | 72.61 | 66.50 | 62.53 | 81.28 ± 0.24 | 70.45 ± 0.60 |
| ✓ | | ✓ | 97.15 | 91.12 | 77.26 | 94.33 | 93.87 | 92.15 | 74.48 | 84.81 | 69.79 | 84.60 | 75.23 | 69.24 | 61.89 | 82.00 ± 0.16 | 71.31 ± 0.16 |
| ✓ | ✓ | ✓ | 96.77 | 93.48 | 83.19 | 94.51 | 93.48 | 92.95 | 75.97 | 84.63 | 71.92 | 85.87 | 75.74 | 71.10 | 67.77 | 83.64 ± 0.42 | 73.08 ± 0.38 |

**Table 4: Quantitative results (mean Dice and mean Jaccard) for the ablation study of the 3D-CGMM training loss. The bold and underlined text denote the best and second best performances, respectively.**

| $\mathcal{L}_{gt}$ | $\mathcal{L}_{self}$ | $\mathcal{L}_{max}$ | $\mathcal{L}_{cons}$ | Mean Dice | Mean Jaccard |
|---|---|---|---|---|---|
| ✓ | | | | 81.29 ± 0.95 | 70.76 ± 1.11 |
| ✓ | ✓ | | | 83.08 ± 0.17 | 72.48 ± 0.23 |
| ✓ | | ✓ | | 83.02 ± 0.21 | 72.29 ± 0.19 |
| ✓ | ✓ | ✓ | | 83.28 ± 0.26 | 72.58 ± 0.46 |
| ✓ | ✓ | ✓ | ✓ | 83.64 ± 0.42 | 73.08 ± 0.38 |

results and achieves a more accurate segmentation than the other methods, demonstrating its effectiveness in accurately segmenting organs in medical images.

## 4.3 Ablation Studies

In our ablation study, we investigate several aspects that can impact the performance of GuidedNet, including each component, the utilization of various loss terms in the 3D-CGMM training loss, and the settings of the hyperparameters. We perform the ablation studies on the FLARE22 dataset with a training set comprised of 42 labeled and 378 unlabeled cases.

**The Effectiveness of Each Component in GuidedNet.** We conduct ablation studies to show the impact of each component in GuidedNet, the results of which are shown in Table 3. For a fair comparison, we evaluate one component per experiment while keeping the others fixed. The first row in Table 3 represents the CPS baseline, on which our method is based. Compared to the baseline, employing the KT-CPS strategy and 3D-CGMM individually yields improvements in the mean Dice of 1.96% and 2.68%, respectively. Additionally, integrating the KT-CPS and 3D-CGMM together yields the highest mean Dice (83.28%) and mean Jaccard (72.58%), outperforming the baseline by 3.96% and 4.44%, respectively. For the segmentation of small and complex organs, the proposed KT-CPS strategy achieves more accurate segmentation results than the baseline. Specifically, the baseline with the KT-CPS strategy achieves a higher Dice for the segmentation of small organs (i.e., LAG, 62.53%; RAG, 66.50%; Gal, 75.61%; and Duo, 70.43%) and complex organs (i.e., Pan, 73.53%; and Sto, 81.53%). When the baseline is combined with the 3D-CGMM, improvements in the Dice are observed for all organs except the Spl and R.kid. This indicates that the 3D-CGMM effectively enhances the quality of pseudo-labels,thereby improving the accuracy of semi-supervised multi-organ segmentation. The individual components employed in our GuidedNet demonstrate standalone benefits, and therefore combining them led to significantly improved optimization.

**Design Choices of 3D-CGMM Training Loss .** Compared to the baseline method that used $\mathcal{L}_{gt}$ only, employing the self-supervision loss $\mathcal{L}_{self}$ and maximization loss $\mathcal{L}_{max}$ individually yields improvements in the mean Dice of 1.79% and 1.73%, respectively. Integrating the self-supervision loss $\mathcal{L}_{self}$ and maximization loss $\mathcal{L}_{max}$ together yields 83.28% for the mean

Dice and 72.58% for the mean Jaccard. Adopting the consistency loss $\mathcal{L}_{cons}$ yields the highest mean Dice (83.64%) and mean Jaccard (73.08%), outperforming the baseline by 2.35% and 2.32%, respectively. The consistency loss $\mathcal{L}_{cons}$ also improved the performances, especially for SSL.

**Ablation Study on HyperParameters.** To validate the robustness of GuidedNet, we conduct ablation studies on the hyperparameters, including the 3D-CGMM training loss weight $\lambda_g$, and a selection of layer features in the decoder used to train the 3D-CGMM. The quantitative results for different hyperparameters are presented in Fig. 6.

**(a) The loss weight $\lambda_g$:** $\lambda_g$ determines the contribution of $\mathcal{L}_{cgmm}$ to the total loss. As shown in Fig. 6(a), $\lambda_g$ = 0.3 results in the highest mean Dice and mean Jaccard.

**(b) Selection of Feature Layers:** Layers 1-4 represent the first to fourth layers of the decoder network (from deep to shallow). Utilizing the features from Layer 4 to model the 3D-CGMM produced the best results, but the effectiveness decreased with shallower layers. This can be attributed to the excessive abstraction in fitting the GMM in high-dimensional space, which may lead to overfitting or instability.

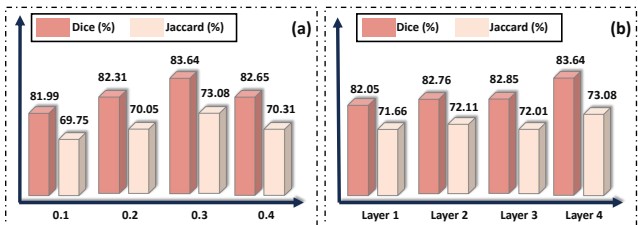

**Figure 6: Quantitative comparisons between different hyperparameters for the FLARE22 dataset: mean Dice and mean Jaccard generated by GuidedNet when trained with various (a) $\lambda_g$ values and (b) layers.**

## 5 CONCLUSION

In this paper, we propose a simple yet effective framework for semi-supervised multi-organ segmentation called GuidedNet. It leverages the knowledge obtained from labeled data to guide the training of unlabeled data, which improves the quality of pseudo-labels for unlabeled data and enhances the network's learning capability for both small and complex organs. Two essential components are proposed: (1) 3D-CGMM models the feature distribution of each class from the labeled data to guide the prediction process for the unlabeled data, and (2) KT-CPS re-weights the pseudo supervised loss terms based on the prior knowledge obtained from the labeled data, forcing the model to pay more attention to organs that are either small or challenging to segment. Comparative experiments against state-of-the-art methods and extensive ablation studies demonstrate the effectiveness of GuidedNet. The visualization results demonstrate that GuidedNet performs well and is effective.

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
