# OpenReview forum: "GuidedNet: Semi-Supervised Multi-Organ Segmentation via Labeled Data Guide Unlabeled Data"
_acmmm.org/ACMMM/2024/Conference — MM2024 Poster_

### Official Review · Reviewer_xdCj · 2024-05-23

**Rating:** 4
**Confidence:** 3

**Summary:**

The paper introduces GuidedNet, a semi-supervised framework for multi-organ segmentation in medical images. The method leverages labeled data to guide the training of unlabeled data, improving pseudo-label quality and segmentation accuracy, particularly for small and complex organs. Key components include:

1. 3D Consistent Gaussian Mixture Model (3D-CGMM): Utilizes feature distributions from labeled data to refine pseudo-labels.
2. Knowledge Transfer Cross Pseudo Supervision (KT-CPS): Uses prior knowledge from labeled data to enhance the training process of unlabeled data.

GuidedNet's effectiveness is demonstrated through extensive experiments on the FLARE22 and AMOS datasets, outperforming state-of-the-art methods in terms of segmentation accuracy.

**Strengths:**

1. Innovative Use of Labeled Data: The integration of 3D-CGMM and KT-CPS uniquely leverages labeled data to guide the segmentation process of unlabeled data, leading to higher accuracy.
2. Enhanced Segmentation for Challenging Organs: Demonstrates significant improvements in segmenting small and complex organs, which are typically difficult for existing methods.
3. Comprehensive Experiments: Extensive evaluations on multiple public datasets (FLARE22 and AMOS) show superior performance over nine state-of-the-art methods.
4. Robust Ablation Studies: Detailed ablation studies highlight the effectiveness of each component and the chosen hyperparameters, reinforcing the robustness of the framework.

**Limitations:**

The narration, drawings and experiments of the whole article are quite reasonable, but I have the following questions:
1. As for the motivation and design of KT-CPS, I know that CPS is an innovative work with good results. Could you please introduce the differences between CPS and KT-CPS in rebuttal?
2. In Table 1 and Table.2, Can you explain why the network does not achieve optimal results in some organs?
3. The combined use of 3D-CGMM and KT-CPS, along with specific hyperparameters, might complicate the implementation and tuning process for practical applications.
4. The GMM model takes a long time to compute, and 3DGMM is even longer. Does it take into account time and storage costs? The approach may face scalability issues when applied to extremely large datasets or when dealing with real-time applications due to its computational requirements.
5. Is any experiments conducted at 1% labeled Settings?

**Suitability:**

2

---

### Official Review · Reviewer_1gvD · 2024-05-23

**Rating:** 4
**Confidence:** 3

**Summary:**

Regarding semi-supervised multi-organ medical image segmentation, the paper proposes a method called GuidedNet, which uses the knowledge in the labeled data to guide the training of unlabeled data, aiming to improve the segmentation quality of unlabeled data. A 3D-CGMM method is proposed, which uses the feature distribution of the labeled data to correct the generated pseudo-labels; a KT-CPS strategy is proposed, which uses the prior knowledge obtained from the labeled data to guide the training of the unlabeled data, and enhances the network's learning ability for small and complex organs. Experiments on FLARE22 and AMOS show that GuidedNet can achieve advanced performance.

**Strengths:**

1. The analysis of the problem is novel and uniquely considers the issue of poorer segmentation effects for small and complex organs in multi-organ medical image segmentation (using the stomach category in FLARE22 as an example).
2. State-of-the-art performance was achieved on the FLARE22 and AMOS datasets.

**Limitations:**

1. The overall innovation of the paper is insufficient. Using labeled data to guide unlabeled data seems more like a combination of several papers (CAML[1], PRCL[2]), and it does not consider the uniqueness of medical organ segmentation using GMM.
2. The authors could consider showing the visualization of the stomach as a complex organ in Figure 1 or Figure 2. How did the authors define the stomach as a complex organ? Maybe it's just because the Dice is poor?
3. The order of Figure 2 could be sorted according to Dice or Euclidean distance, which would present a better visualization.
4. The representations in lines 1, 3, 4, and 13 of Algorithm 1 can be more accurate. The result can only be obtained after $x_i$ goes through the model.
5. In 3D-CGMM, I hope the authors can provide relevant explanations: What are the advantages of modeling the feature space in the form of Gaussian distribution compared to Prototype and probabilistic representation?
6. The definition given by KT-CPS in the formula is to add a weight mechanism in CPS cross supervision, but the direction of knowledge in Figure 3 (from the green feature map to the blue feature map) is not clear enough.
7. The weight mechanism used in KT-CPS is the ratio of voxel numbers, and the paper lacks an explanation of the role of KT-CPS in dealing with complex organs.
8. The experiment lacks ablation of $L_{rectify}$.
9. The paper needs to add ablation experiments for the hyperparameter $\lambda_u$.

Reference

[1] Gao S, Zhang Z, Ma J, et al. Correlation-aware mutual learning for semi-supervised medical image segmentation[C]//International Conference on Medical Image Computing and Computer-Assisted Intervention. Cham: Springer Nature Switzerland, 2023: 98-108.

[2] Xie H, Wang C, Zheng M, et al. Boosting semi-supervised semantic segmentation with probabilistic representations[C]//Proceedings of the AAAI Conference on Artificial Intelligence. 2023, 37(3): 2938-2946.

**Suitability:**

2

---

### Official Review · Reviewer_eMYW · 2024-05-24

**Rating:** 3
**Confidence:** 2

**Summary:**

This paper introduces a 3D Consistent Gaussian Mixture Model (3D-CGMM) tailored to utilize feature distributions from labeled data to refine generated pseudo-labels. Additionally, it proposes a Knowledge Transfer Cross Pseudo Supervision (KT-CPS) strategy, which leverages prior knowledge gleaned from labeled data to enhance the training of unlabeled data. This approach boosts segmentation accuracy for both small and intricate organs.

**Strengths:**

This paper is well written and the figures are great

**Limitations:**

Making the information of labeled and unlabeled data interact with each other during the training process is not novel for semi-supervised medical image segmentation. The authors should add more explanation to interpret why the proposed method is different with these relavant models like BCP [1]. Moreover, the proposed model should compare with these models.
[1] Y. Bai, D. Chen, Q. Li, W. Shen, and Y. Wang, “Bidirectional copy-paste for semi-supervised medical image segmentation,” in CVPR, pp. 11514–11524, 2023.

**Suitability:**

2

---

### Meta-Review · Area_Chair_ULKR · 2024-07-09

**Recommendation:** Accept (Poster)
**Confidence:** 5

**Metareview:**

All the reviewers agreed to accept this paper after rebuttal. I, therefore, would recommend to accept it.